# Docetaxel Resistance in Castration-Resistant Prostate Cancer: Transcriptomic Determinants and the Effect of Inhibiting Wnt/β-Catenin Signaling by XAV939

**DOI:** 10.3390/ijms232112837

**Published:** 2022-10-25

**Authors:** Elena Pudova, Anastasiya Kobelyatskaya, Irina Katunina, Anastasiya Snezhkina, Kirill Nyushko, Maria Fedorova, Vladislav Pavlov, Elizaveta Bulavkina, Alexandra Dalina, Sergey Tkachev, Boris Alekseev, George Krasnov, Vsevolod Volodin, Anna Kudryavtseva

**Affiliations:** 1Engelhardt Institute of Molecular Biology, Russian Academy of Sciences, 119991 Moscow, Russia; 2National Medical Research Radiological Center, Ministry of Health of the Russian Federation, 125284 Moscow, Russia; 3Institute for Regenerative Medicine, Sechenov First Moscow State Medical University (Sechenov University), 119991 Moscow, Russia

**Keywords:** CRPC, RNA-Seq, docetaxel, resistance, XAV939, transcriptomics, expression, pathways, mRNA isoforms, exosomes

## Abstract

Castration-resistant prostate cancer (CRPC) is a common form of prostate cancer in which docetaxel-based chemotherapy is used as the first line. The present study is devoted to the analysis of transcriptome profiles of tumor cells in the development of resistance to docetaxel as well as to the assessment of the combined effect with the XAV939 tankyrase inhibitor on maintaining the sensitivity of tumor cells to chemotherapy. RNA-Seq was performed for experimental PC3 cell lines as well as for plasma exosome samples from patients with CRPC. We have identified key biological processes and identified a signature based on the expression of 17 mRNA isoforms associated with the development of docetaxel resistance in PC3 cells. Transcripts were found in exosome samples, the increased expression of which was associated with the onset of progression of CRPC during therapy. The suppression of pathways associated with the participation of cellular microtubules has also been shown when cells are treated with docetaxel in the presence of XAV939. These results highlight the importance of further research into XAV939 as a therapeutic agent in the treatment of CRPC; moreover, we have proposed a number of mRNA isoforms with high predictive potential, which can be considered as promising markers of response to docetaxel.

## 1. Introduction

Prostate cancer (PCa) is one of the most urgent problems of modern oncology and is characterized by high incidence rates among men worldwide [1]. The main method of treatment for patients with advanced stages of PCa is androgen deprivation. However, after temporary stabilization, the majority of patients in whom progression of the tumor process is observed against a background of castration testosterone levels pass into the stage of castration-resistant PCa (CRPC). Metastatic CRPC is an extremely unfavorable form of the disease, which significantly impairs the quality and reduces the life expectancy of patients [2].

The standard treatment for patients with metastatic CRPC is chemotherapy with cytotoxic drugs from the taxane class. The action of taxanes is to polymerize tubulin into stable microtubules, which leads to the arrest of cell division. However, an inevitable development of resistance to taxanes eventually occurs in the patients, this presenting a serious problem in modern oncourology. Here, we highlight several important areas of research aimed at solving it [3,4].

First is the identification of key molecular genetic changes and informative markers of response to taxane therapy. In recent years, the relevance of the search for promising markers based on the expression of various molecules, both in tumor cells and in biological fluids using the liquid biopsy approach, has been noted [5,6,7]. The use of high-throughput sequencing technology makes it possible to carry out large-scale profiling of the expression of genes and regulatory molecules, such as lncRNAs and microRNAs, both in tumor samples and in tumor exosomes, and this is opening up new frontiers in the search for promising markers and therapeutic targets [8,9]. Also, with the appearance of the RNA-Seq approach, it became possible to study more closely the phenomenon of alternative splicing, a process that reflects an additional level of gene regulation and is the main mechanism that ensures proteome diversity. The formation of different protein isoforms with different biological functions occurs as a result of rearrangement of the coding and non-coding regions of a particulat gene, with the formation of several mRNA transcript variants. A number of studies have shown a relationship between aberrant mRNA isoforms and the development of cancer, as well as enabling a more informative separation of normal and tumor cells based on the expression of such mRNA isoforms, and this includes the case of PCa [10,11].

This paper describes our search for promising therapeutic agents that, in combination with taxanes, can prolong the sensitivity of tumor cells to chemotherapy. It is known that one of the key biological processes closely associated with the development, progression, and resistance to PCa therapy is the canonical Wnt/β-Catenin signaling pathway [12]. In metastatic CRPC, increased expression of nuclear β-catenin and the target genes of the Wnt/β-catenin signaling pathway have been shown. It is thought that this signaling pathway underlies the acquisition of a selective advantage by cells under the influence of therapy [13,14]. One promising agent for study targeting Wnt/β-catenin signaling is the tankyrase inhibitor XAV939. It has been noted that the inhibition of Wnt/β-catenin signaling in tumor cells can increase their sensitivity to radio-, chemo-, or hormone therapy or lead to impaired proliferation, depending on the concentration of XAV939 [15,16,17,18,19].

The present study is devoted to an analysis of transcriptomic profiling of the experimental cell line PC3, in the process of acquiring resistance to docetaxel, using the RNA-Seq approach. The work will consider the main biological processes and differentially expressed mRNA isoforms potentially associated with the development of such resistance to docetaxel. Also, the effect of XAV939 in combination with docetaxel at the transcriptomic level will be evaluated for PC3 cells. Given the relevance of the search for predictive markers based on minimally invasive liquid biopsy, the study included transcriptomic profiling of blood plasma exosomes in patients with CRPC during docetaxel therapy. The results obtained contribute to our understanding of the key processes underlying the development of resistance of CRPC tumor cells to chemotherapy with docetaxel and highlight key transcripts based on mRNA expression, that offer a high predictive potential for use in further studies.

## 2. Results

### 2.1. Analysis of the Biological Pathways Associated with Docetaxel Resistance

Based on RNA-Seq data the for experimental PC3 cells after each round of docetaxel treatment (4 nM, 8 nM, and 10 nM concentrations), we obtained lists of differentially expressed genes (DEGs), both when comparing treated cells with their respective temporal controls and when switching cells into a docetaxel-resistant state (Dox 8 nM vs. Dox 10 nM). Full lists of DEGs for each comparison are presented in Appendix A. Pathway enrichment analysis was performed for the resulting DEGs lists using the GSEA algorithm. The main results of the pathway analysis for each comparison are presented in Table 1.

Considering the results obtained in this comparison, we see an increase in the activation of pathways associated with inflammation, which is quite understandable as a response of cells to an increase in the concentration of docetaxel. A further interesting observation is the activation of pathways associated with lipid metabolism (GO:0008203, GO:0055092, GO:0006695, GO:0019216).

The RNA-Seq data from the PRJNA589746 project for PC3 and DU145 cells were also analyzed to compare results. When performing such a comparison for the results of pathway enrichment analysis, statistically significant intersections of the pathways GO:0014070, GO:0050728, GO:0006694 and GO:0034599 were found (Table 2).

It was noted that for all datasets, the “negative regulation of inflammatory response” (GO:0050728) pathway was upregulated with the development of docetaxel resistance.

### 2.2. Transcriptome Analysis of Plasma Exosomes in CRPC Patients during Docetaxel Therapy

We obtained RNA-Seq data on the complete transcriptome of blood plasma exosomes of Russian patients with CRPC treated with docetaxel. Baseline DE analysis was performed between treatment response and conditional progression (significant rise of prostate specific antigen) samples in patients in pairwise comparisons. As a result, we obtained a list of statistically significant DEGs (Appendix A).

Taking into account the fact that exosomes can reflect the transcriptomic pattern of tumor cells, we performed an over-representation analysis (ORA) to determine the biological pathways in which the identified genes are involved. Representation analysis was performed for list of overexpressed genes in early treatment resistance samples (Figure 1).

This analysis mainly identified pathways associated with inflammation. There are also a number of pathways that downregulate apoptosis in response to DNA damage controlled by the p53 class mediator.

### 2.3. Features of the PC3 Cells Transcriptome Profile after Treatment with XAV939

We obtained RNA-Seq data after each round of cell treatment with XAV939 at 2.5 µM. Previously, we had noted a toxic effect of XAV939 at this concentration after three cycles of treatment of the PC3 cells compared with the control where no such effect was detected (Figure 2a). As a result of the DE analysis, relative to the control points of similar passages, lists of DEGs were obtained. The pattern of distribution of DEGs after each cell treatment is shown in Figure 2b. Full lists of DEGs for each comparison are presented in Appendix A. Pathway enrichment analysis using the GSEA algorithm for each cell treatment compared to the controls did not reveal statistically significant biological pathways (FDR < 0.05).

### 2.4. Effect of the XAV939+Docetaxel Combination on the PC3 Cells Transcriptome

RNA-Seq data were obtained after each round of cell treatment with various concentrations of docetaxel in combination with XAV939. As a result of the DE analysis in comparison with the controls, lists of DEGs were obtained. Full DEGs lists for each comparison are presented in Appendix A.

Next, we conducted pathway enrichment analysis. As a result of our analysis, we focused on pathways that were statistically significantly altered when the cells were treated with combinations of XAV939+DOX 8 nM and XAV939+DOX 10 nM. Figure 3 shows a heat map with the main results.

According to the results of the analysis, we see that in the case of the comparisons under consideration, activation of inflammation pathways is preserved, this corresponding to the response of cells to the administered taxane. The most interesting result of this analysis is the suppression of pathways associated with cell mitosis, especially those involving microtubules. Thus, we see that during the prolonged treatment of PC3 cells with docetaxel in combination with XAV939, chemosensitivity to docetaxel is preserved at the transcriptomic level, when compared with the PC3 control cells.

Separately, we reviewed the pathways altered when comparing PC3 cells after XAV939+DOX 8 nM and 10 nM treatment (Table 3). Considering a similar comparison without XAV939, where all statistically significant pathways were activated and predominantly associated with inflammation from docetaxel exposure, we see only one similarly activated pathway. Most of the identified biological pathways are suppressed in this case, being associated with DNA repair, transcription, translation, and related processes.

### 2.5. Identification of Promising lncRNAs Associated with Docetaxel Resistance

One of the goals of our study was to identify potential markers of response to docetaxel therapy, in particular at the level of lncRNAs expression. We identified a number of lncRNAs whose expression changes statistically significantly in the resulting DE lists for plasma exosome data, experimental PC3 cells data, and PRJNA589746 project data (Table 4).

We identified five lncRNAs whose DE was statistically significant found to be unidirectional in most of the considered datasets. It should be noted that among the considered lncRNAs, DE of *MIR222HG* was found in both PC3 cells and in blood plasma exosomes.

### 2.6. DE Transcripts as Potential Markers of Response to Docetaxel Based on PC3 Cell Data

To search for potential markers of response to docetaxel between the selected periods of docetaxel treatment of PC3 cells, we analyzed the DE of gene isoforms. The main criterion for filtering the results was the threshold of statistical significance of the QLF test (FDR < 0.05). We also crossed the obtained list of isoforms with the results of DE of conditionally resistant PC3 cells relative to the control (QLF FDR < 0.05). Complete lists of DE isoforms of genes are presented in Appendix A. Table 5 shows the found DE transcripts.

For the identified protein-coding mRNA isoforms, their protein products were also determined. The results are presented in Table 6.

We also assessed the DE of signature participants at the gene level using data from the PRJNA589746 project (Table 7). Statistically significant results in the case of PC3 cells were confirmed based on the expression of the *PRSS3*, *FTL*, *UBC*, *SQSTM1* and *CFL1* genes. It is also worth noting that the expression of the *PRSS3* gene also increased during the development of resistance to docetaxel in the case of DU-145 cells.

### 2.7. Evaluation of the Effect of XAV939+Docetaxel Combination on PC3 Cells Based on the Identified DE Transcripts

Previously, we found a transcriptomic signature based on 17 DE gene isoforms, which potentially reflects the development of PC3 cells resistance to docetaxel. Considering this transcript list, we also evaluated the effect of adding XAV939 to PC3 cells in a similar comparison with docetaxel (XAV939+DOX 8 nM vs. XAV939+DOX 10 nM) (Appendix A). The result is shown in Figure 4. It was found that in the case of the combination, almost all transcripts showed a decrease in expression (Table 8).

In addition, it is worth noting that the DE of MIR222HG in the case of PC3 cells are treated with XAV939+Docetaxel combination also increases (Log2FC = −0.1).

### 2.8. DE Transcripts as Potential Markers of Docetaxel Response Based on the Exosome Plasma Samples

The DE analysis was also performed in the gene isoform evaluation mode; statistically significant results are collected in Appendix A.

Considering the DE analysis of isoforms, we were mainly interested in transcripts of the ‘lncRNA’ type. After filtering the list of DE isoforms by *p*-value QLF and Spearman’s rank correlation, four transcripts were identified that meet the specified criteria: *RPL7A-205*, *NOP53-207*, *CAPZA1-204*, and *MALAT1-201*. Expression data are presented in Table 9.

We also reviewed the expression of these transcripts in resistant PC3 cells. It was found that the differential expression of *RPL7A-205* and *CAPZA1-204* transcripts was also increased and was statistically significant (Log2FC = 6.9; QLF *p*-value = 0.01 and Log2FC = 3.17; QLF *p*-value = 0.005, respectively).

### 2.9. Validation of Docetaxel Resistance Potential Markers Expression by Quantitative PCR

We have validated the relative expression of a number of transcripts. In the case of experimental PC3 cells, *AMBRA1-213*, *CFL1-201* and *TUBB3-201* transcripts were selected. The expression of these transcripts increases more than 2-fold with the development of resistance to docetaxel and changes most strongly when cells are treated with docetaxel in combination with XAV939. The relative expression of *MIR222HG* was also validated.

During the validation of the selected transcripts, it was found that the AMBRA1-213 transcript had low expression and was subsequently excluded from the analysis. The statistical significance of the obtained results of transcript expression between groups was assessed using a t-test (Figure 5 and Table 10).

Based on the relative expression of MIR222HG, CFL1-201, and TUBB3-201, a statistically significant difference between DOX 4 nM vs. DOX 10 nM and Control 3 vs. DOX 10 nM was confirmed (*p*-value < 0.05). Based on the expression of MIR222HG and CFL1-201, a statistically significant difference was found between the DOX 8 nM and DOX 10 nM groups.

We also validated the relative expression of MIR222HG, RPL7A-205, and CAPZA1-204 transcripts on plasma exosome samples from patients with CRPC. The choice of RPL7A-205 and CAPZA1-204 transcripts for validation was determined by Log2FC > 3. The statistical significance of the obtained results of transcript expression between periods of response to therapy and progression in patients was assessed using a paired Wilcoxon test (Figure 6).

It was found that a statistically significant difference between the considered groups is observed only on the basis of the relative expression of MIR222HG (*p*-value = 0.001).

## 3. Discussion

Taxanes are a class of antimitotic drugs that stabilize cellular microtubules by reversibly binding to the β-subunit of the tubulin heterodimer. Microtubules mediate a variety of cellular processes that are key to tumor cell proliferation and metastasis. Microtubules are characterized by “dynamic instability”, which provides an optimal change in the phases of growth and shortening during cell division. The action of taxanes in the framework of microtubule polymerization causes the formation of stable bundles, which, as a result, changes their dynamics, blocks mitosis, and promotes apoptosis [20]. The most common drugs in this class include paclitaxel, cabazitaxel and docetaxel; the latter, in turn, is used as the main chemotherapy drug for the treatment of CRPC. Tubulin polymers formed under the actions of docetaxel and paclitaxel have structural differences. In the case of the action of docetaxel, the resulting polymers are more efficient due to the binding to β-tubulin with greater affinity. Docetaxel has also been shown to be highly efficacious compared to paclitaxel, resulting in the widespread use of this taxane [21].

Chemotherapy with docetaxel improves the survival of patients with CRPC, but half of the patients begin to stop responding to cytotoxic treatment. The development of resistance to taxanes involves many mechanisms and genes that work alone or in combination with other factors to inhibit their action. Considering various studies, some mechanisms associated with resistance to taxanes have been identified, but the full picture of this phenomenon remains to be elucidated. The most well-known processes associated with the development of resistance to docetaxel include the following: structural and functional changes in microtubules (βIII-tubulin overexpression, mutations in β- or α-tubulin genes), upregulation of the drug efflux transporter (activation of members of the ABC transporters family), activation of survival pathways/escape from apoptosis (activation of anti-apoptotic proteins of the BCL2 family, upregulation of *GATA2*, activation of the PI3K/AKT, Notch, Hedgehog, JNK and WNT/β-Catenin pathways), altered AR signaling, activation of antioxidant response (high expression of BIM-1, SOD2 and IGF-1 signaling), hypoxia, non-coding RNAs, and others [22,23,24,25,26].

In the present study, we reviewed the main transcriptomic changes associated with the development of docetaxel resistance, as well as the effect on maintaining sensitivity to docetaxel using the XAV939 agent based on the experimental cultivation of the PC3 cell line. Utilizing RNA-Seq profiling and subsequent bioinformatic analysis, we analyzed enriched biological pathways associated with the development of docetaxel resistance. The analysis showed the activation of pathways associated with inflammation as well as signaling through cytokines. These processes are involved in all stages of PCa progression, including resistance to cytotoxic drugs in the case of CRPC. The interaction between tumor cells and neighboring myeloid cells by cytokine mediators is important for the development of some forms of chemoresistance [27]. Researchers have shown that chemotherapy resistance, both in vitro and in patients, is associated with an inflammatory response involving cytokines associated with macrophage recruitment and activation. The addition of docetaxel to PC3-U937 co-culture increases cytokine production. In addition, correlation studies in humans have identified a circulating cytokine profile that is associated with treatment response in patients with CRPC [28]. Statistically significant activation of the inflammation category pathway (GO:0050728) in the development of docetaxel resistance was also shown by comparing the results for experimental PC3 cells with data from the PRJNA589746 project for PC3 and DU-145 cell lines.

In addition, according to the results of the analysis, we observed the activation of pathways associated with lipid metabolism. Aberrant lipid metabolism is one of the new characteristics of the development of chemoresistance of tumor cells, and a complete picture of this mechanism remains to be elucidated. It is known that lipid biosynthesis is controlled by many different regulatory networks and contributes to the progression of cancer. Currently, it is known that one of the key regulators of lipogenesis in tumor cells is proteins that bind sterol regulatory elements (SREBP), which in turn transcriptomically activate such enzymes as ATP-citrate lyase (ACLY), acetyl-CoA carboxylase (ACC), fatty acid synthase (FASN), and acyl-CoA synthetase (ACS) [29]. The potential involvement of these enzymes in the interaction between lipogenesis and chemoresistance has been shown [30,31]. Thus, the features of lipid metabolism and the expression of its participating genes during the progression of prostate cancer, including the development of drug resistance, are of increasing interest for in-depth study.

Additionally, transcriptome profiling of plasma exosome samples from patients with CRPC treated with docetaxel was performed. A special focus of the study was directed to the identification of transcripts, especially lncRNA, whose expression is upregulated in exosomes during disease progression. Based on the obtained total gene lists, we performed an overrepresentation analysis (ORA) to determine the association of gene sets with biological pathways. We have shown that in addition to the background inflammation from docetaxel, the expression of genes involved in the negative regulation of apoptosis in exosomes increases in response to DNA damage controlled by the p53 class mediator. The action of docetaxel on tumor cells, after the polymerization of microtubules, causes mitotic catastrophe and caspase-2 and -3-dependent apoptosis [32,33]. Defects in apoptosis signaling pathways can lead to the development of drug resistance, which ultimately limits the effectiveness of therapy. Thus, signaling pathways for apoptotic cell death can be considered as one of the promising therapeutic strategies to increase the effectiveness of chemotherapy.

Thus, comparing the results obtained from the data of samples of PCa cell lines and the blood plasma exosomes of patients with CRPC during the development of resistance to docetaxel showed a significant increase in the expression of genes involved in the biological pathways responsible for inflammation.

When evaluating pathway enrichment in the case of docetaxel treatment of PC3 cells in combination with XAV939, we also observed a retention of inflammatory pathway activation as well as significant suppression of microtubule-mediated cell division biological processes. Moreover, this effect was also preserved during the subsequent treatment of cells with docetaxel at a concentration of 10 nM. Thus, it can be assumed that under the influence of XAV939, PC3 cells retain sensitivity to docetaxel, which manifests itself in the preservation of the main cytostatic effect of this taxane.

As part of this study, we also considered lncRNAs, the expression of which may potentially reflect the development of resistance to docetaxel. LncRNAs are a special class of regulatory RNA molecules that perform a variety of functions, such as epigenetic regulation and gene regulation in tumor cells, including prostate cancer. In various types of cancer, lncRNA can act as both tumor suppressors and oncogenes, and their involvement in the development and progression of urological tumors has been repeatedly demonstrated [34]. In addition, the development of drug resistance in tumor cells can also occur due to the transfer of specific lncRNA in extracellular vesicles from resistant tumor cells [35]. We identified a number of lncRNAs whose expression was statistically significantly associated with the development of resistance to docetaxel based on both cell line data and plasma exosome data from patients with CRPC. We did not find lncRNAs whose intersections were found in all datasets. However, a number of lncRNAs were isolated that were displayed in most of the data: *AL157938.2, LINC02582, SNHG1, KCNQ1OT1*, and *MIR222HG.*

Among these lncRNAs that significantly and unidirectionally change during the development of resistance to docetaxel in all considered data sets, we specifically examined *MIR222HG*. According to the literature, the expression of this lncRNA is associated with the development of the CRPC phenotype, and it also promotes the androgen-independent growth of the LNCaP cell line [36]. *MIR222HG* was also isolated by researchers as part of the immune-related lncRNA signature associated with an unfavorable prognosis in glioblastoma, including in the composition of exosomes [37]. According to the results of our study, we showed that a decrease in *MIR222HG* expression was statistically significantly associated with the development of resistance to docetaxel both in the case of PC3 cells and in exosomes of patients with CRPC. When comparing PC3 cells after treatment with docetaxel at a concentration of 8 nM versus 10 nM in combination with XAV939, *MIR222HG* expression in the latter increased. The statistical significance of the association of *MIR222HG* expression with the development of resistance to docetaxel was confirmed by qPCR in both PC3 cells samples and plasma exosome samples from patients with CRPC.

We also considered potential markers of the development of resistance to docetaxel in the case of PC3 cells based on the expression of mRNA isoforms: *PRSS3-204, FTL-201, ICAM1-201, RPL28-205, AMBRA1-213, GDF15-201, RPS6-204, IER3-201, RPL10-210, UBC-201, SQSTM1-202, JUNB-201, CFL1-201, TUBB4B-201, TUBB-205, S100A6-201* and *TUBB3-201*.

In this signature, we observe upregulation of the three tubulin-associated transcripts *TUBB4B-201, TUBB-205*, and *TUBB3-201*, the latter more than doubling in resistant cells compared to cells after the previous docetaxel treatment step. The association of *TUBB3* gene overexpression with the development of docetaxel resistance in PCa has also been shown in other studies. In addition, studies have demonstrated that *TUBB3* knockdown resensitized cells resistant to docetaxel [38,39].

Identified transcripts with the most elevated expression include *AMBRA1-213* and *SQSTM1-202* (up to 29 and 2.7 times, respectively), the protein products of which are participants in autophagy. Autophagy in the case of advanced cancers may provide tumor cells with a survival advantage under various stressful conditions, including cytotoxic drugs [40,41]. The pro-autophagic protein *AMBRA1* is involved in the formation of autophagosomes. The p62 protein (SQSTM1) confers autophagy selectivity, playing a critical role in recognition/loading of cargo into autophagosomes. It has been shown that the increased expression of *AMBRA1* in PCa has a significant correlation with Gleason score, and the expression of *AMBRA1* and *SQSTM1* mRNA is significantly upregulated in PCa samples compared to benign prostatic hyperplasia, which may potentially contribute to PCa progression [42]. It has also been shown that *AMBRA1* prevents apoptosis in prostate cancer cells and enhances their colony formation, resulting in resistance to cisplatin [43].

Based on our results, we also observed an increase in the expression of the *GDF15-201* transcript, which was up to 5-fold in docetaxel-resistant cells. Macrophage inhibitory cytokine-1 (MIC-1, growth differentiation factor 15 GDF15) is a member of the TGF-β superfamily and is expressed in many cells, including PCa cells. *GDF15* expression is elevated in tumor cells and increases with progression, especially in CRPC [44]. The overexpression of GDF15 increases the resistance of PCa cells to docetaxel, and when *GDF15* is suppressed by siRNA, an increase in sensitivity to docetaxel is observed [45].

It is also worth noting another member of this signature, *CFL1-201*. The *CFL1* gene encodes a protein that plays a key role in cell migration and cytokinesis and is regulated by factors such as phosphorylation, pH, phospho-inositide binding, and subcellular compartmentalization [46]. This protein has been reported to be directly associated with invasion, metastasis, and chemoresistance in various types of cancer [47,48]. In the case of PCa, it has been shown that CFL1 expression in the mesenchyme may be closely associated with the development of lymph node metastases [49].

The effect of the XAV939+DOX combination was also estimated based on the identified transcriptome signature. We reviewed signature expression when comparing PC3 cells after treatment with 8 nM and 10 nM docetaxel in combination with XAV939, and we found that all transcripts showed downregulation.

We also validated the expression of *CFL1-201* and *TUBB3-201* transcripts from this signature, the differential expression of which changed most significantly both in the case of cells exposed to docetaxel (Log2FC > 2), and it significantly decreased against the background of XAV939. As a result of the validation, the statistical significance of the expression of these transcripts with the development of resistance to docetaxel was confirmed.

## 4. Materials and Methods

### 4.1. PC3 Cells

The PC3 cell line was obtained from the American Tissue Culture Collection (ATCC, Manassas, VA, USA). Cell cultivation was carried out in RPMI-1640 medium supplemented with 10% FBS, L-glutamine, 100 U/mL penicillin, and 100 mg/mL streptomycin at 37 °C with 5% CO_2_.

### 4.2. Treatments of PC3 Cells

Obtaining a resistant PC3 subline was carried out based on a stepwise increase in the concentration of taxane docetaxel (Sigma Aldrich, Burlington, MA, USA) in culture medium with concentrations of 4 nM, 8 nM and 10 nM. Cells that survived after cultivation at a docetaxel concentration of 10 nM were considered conditionally resistant. To inhibit the Wnt signaling pathway, XAV939 (Sigma Aldrich) was used in a culture medium at a concentration of 2.5 µM, which was selected on the basis of literature data.

### 4.3. Isolation of Total RNA from Cells

The isolation of total RNA from cells was performed using the RNeasy Micro Kit (Qiagen, Hilden, Germany) according to the manufacturer’s protocol. The concentration of the obtained total RNA was estimated using the Qubit™ RNA BR Assay Kit (Thermo Fisher Scientific, Waltham, MA, USA) on a Qubit 4.0 fluorometer according to the manufacturer’s protocol. The quality of the obtained total RNA (RNA Integrity Number; RIN) was assessed using the Agilent RNA 6000 Nano Kit (Agilent Technologies, Santa Clara, CA, USA) on an Agilent 2100 Bioanalyzer. The RIN parameter for the samples was at least 9.

### 4.4. Plasma of Patients with CRPC Treated with Docetaxel

The study included 20 plasma samples from 10 Russian patients with metastatic CRPC during chemotherapy with docetaxel, which were obtained under observation in the National Medical Research Radiological Center, Ministry of Health of the Russian Federation (Table 11). The majority of patients included in the study had previously undergone radical prostatectomy. Samples were divided into two groups during the observation process in accordance with the dynamics of the prostate-specific antigen (PSA) level in each patient during treatment: response to therapy (stable low PSA level of the patient), and progression (elevated PSA level in a patient). Plasma sample preparation, hemolysis assessment, and PSA measurement were performed as previously described [50].

### 4.5. Isolation of Total Exosomal RNA from Blood Plasma Samples

Blood plasma samples were subjected to additional purification through specialized filters with a pore size of 0.8 µm (Sartorius, Göttingen, Germany). The isolation of total exosomal RNA was performed from 1 mL of filtered blood plasma using the exoRNeasy Serum-Plasma Midi Kit (Qiagen, Hilden, Germany) according to the manufacturer’s protocol. The concentration of the obtained total RNA was estimated using the Qubit™ RNA HS Assay Kit (Thermo Fisher Scientific, Waltham, MA, USA) on a Qubit 4.0 fluorometer according to the manufacturer’s protocol.

### 4.6. Library Preparation and High Throughput Sequencing

Library preparation based on plasma samples and cell cultures was performed using the KAPA RNA HyperPrep Kit with RiboErase (HMR or Human/Mouse/Rat) (Roche), according to the manufacturer’s protocol. The quality of the resulting transcriptome libraries was analyzed on an Agilent Bioanalyzer 2100 instrument (Agilent Technologies, Santa Clara, CA, USA) using the Agilent High Sensitivity DNA Kit (Agilent Technologies). The concentration of the resulting libraries was measured on a Quibit 4.0 fluorimeter (Thermo Fisher Scientific) using the Qubit dsDNA HS Assay Kit (Thermo Fisher Scientific). The size of the resulting mRNA libraries was ≈500 bp. Sequencing of transcriptome libraries was performed on a NextSeq 2000 instrument using NextSeq 2000 P2 Reagent kits (100-cycle) in single-end read mode.

### 4.7. Bioinformatics Analysis

For the obtained RNA-Seq data in the fastqc format, quality assessment was performed using the FastqQC and MultiQC programs (https://www.bioinformatics.babraham.ac.uk/projects/fastqc/, accessed on 10 May 2022). The Trimmomatic tool was used to remove adapter sequences from RNA-Seq data, which was followed by mapping to the reference genome (GRCh38 assembly) using the STAR tool [51,52]. Differential gene expression analysis was performed in the R statistical environment (v.3.6.3, Vienna, Austria) using the edgeR package (v.3.24.3, Parkville, Australia) [53]. The TMM (Trimmed Mean of M-values) method was used to normalize the data. FeatureCounts (Subread package v.1.6.4, Parkville, Australia) was used to calculate the read counts per gene [54]. The RSEM tool was used to identify transcripts of gene isoforms [55]. Heat map visualization of transcriptome profiles was performed using the ggplot2 and bioinfokit packages [56]. Path enrichment analysis was performed using the library in a Python environment using the gseapy library and the Gene Ontology Biological Process 2021 database. In the analysis of differential gene expression, the quasi-likelihood F-test (QLF test) was used. The Benjamini–Hochberg correction was applied to calculate the false positive rate (FDR). Spearman’s rank correlation coefficient (r_s_) was used for correlation analysis of the data. Differences in the level of gene expression were considered statistically significant at test *p*-values < 0.05.

As part of the bioinformatics analysis, RNA-Seq data from the PRJNA589746 project were also analyzed. Data include docetaxel-sensitive and resistant PC3 (PC3-SC) and DU-145 (DU145-SC) cell lines [57].

### 4.8. Quantitative PCR (qPCR)

cDNA was obtained from total RNA using the Mint kit (Evrogen, Moscow, Russia) according to the manufacturer’s protocol. For PC3 cell samples, qPCR was performed on an Applied Biosystems 7500 instrument (Thermo Fisher Scientific, USA) in three technical replicates. HPRT1 was used as a control gene. For blood plasma exosome samples, qPCR was performed on a Rotor-Gene Q instrument (Qiagen) in three technical replicates. GAPDH was used as a control gene. The level of relative mRNA expression for each comparison was calculated by the ΔCT method. Visualization and statistical analysis of expression results were performed using t and paired Wilcoxon tests in Jupyter Notebook, Python (ver. 3.6, Python Wilmington, DE, USA).

## 5. Conclusions

We performed large-scale RNA-Seq profiling of PC3 cells gradient-treated with docetaxel as well as the combination of docetaxel with the Wnt/β-catenin signaling pathway inhibitor XAV939. RNA-Seq profiling of blood plasma exosomes in patients with CRPC during docetaxel therapy was also performed. Based on the bioinformatics analysis, it was shown that with the development of resistance to docetaxel, the expression of genes participating in the biological pathways of the inflammation category increases. A number of transcripts have been identified whose expression is potentially associated with the development of docetaxel resistance: *PRSS3-204, FTL-201, ICAM1-201, RPL28-205, AMBRA1-213, GDF15-201, RPS6-204, IER3-201, RPL10-210, UBC-201, SQSTM1-202, JUNB-201, CFL1-201, TUBB4B-201, TUBB-205, S100A6-201*, TUBB3-201 and *MIR222HG*. Based on the analysis of biological pathways and transcriptome signature, the effect of preserving the sensitivity of PC3 cells to docetaxel in the presence of XAV939 was shown. A statistically significant association of *MIR222HG* expression with the development of resistance to docetaxel was confirmed by qPCR in both PC3 cells and plasma exosomes of patients with CRPC. The statistically significant expression of *TUBB3-201* and *CFL1-201* transcripts in the development of docetaxel resistance in PC3 cells was also confirmed.

## Figures and Tables

**Figure 1 ijms-23-12837-f001:**
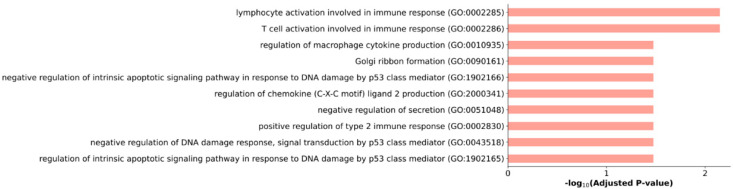
Biological processes involving DE genes in blood plasma exosomes of patients with CRPC during progression of docetaxel therapy. The distribution of pathways is plotted against the level of static significance (−log10 adjusted *p*-value).

**Figure 2 ijms-23-12837-f002:**
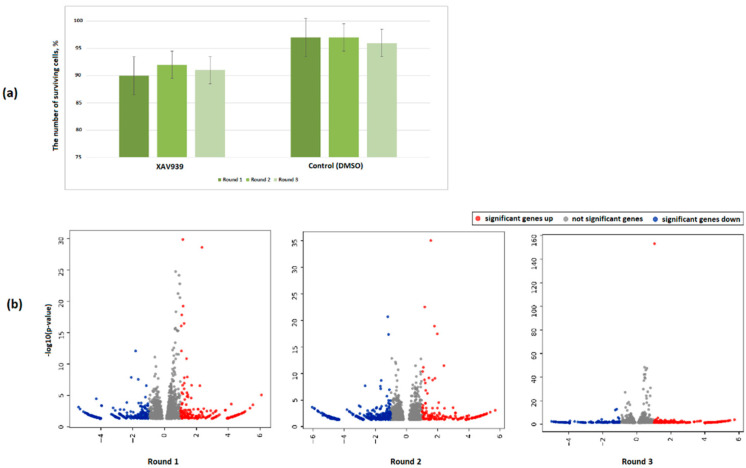
(**a**) The percentage of surviving cells after each round of treatment. The barplots colors reflect the rounds of cell treatments. Cells of similar passages without treatment were taken as 100%. For each round of cell treatment, the average of three experimental replicates is shown. (**b**) Volcano plot of differential gene expression between PC3 cells with sequential XAV939 at 2.5 µM treatments and controls.

**Figure 3 ijms-23-12837-f003:**
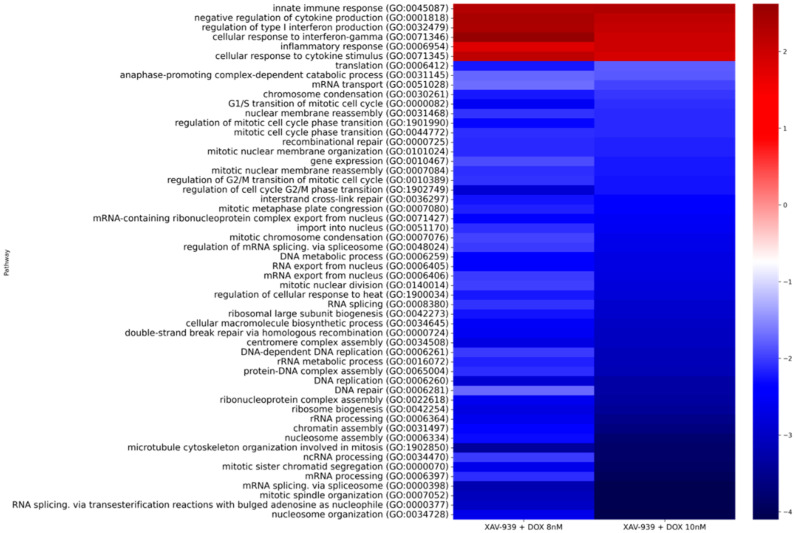
Heat map showing overall results of pathway enrichment analysis according to the Gene Ontology Biological Process 2021 database for PC3 cells treated with XAV939+DOX 8 nM and XAV939+DOX 10 nM compared to control PC3 cells.

**Figure 4 ijms-23-12837-f004:**
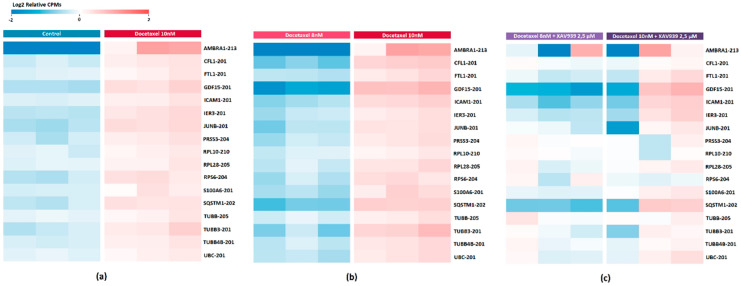
Heat map showing the differential expression of transcripts associated with the development of resistance to docetaxel treatment on PC3 cells (**a**) compared to the control (**b**) after 8 nM versus 10 nM docetaxel treatment (**c**) after 8 nM versus 10 nM docetaxel treatment in combination with XAV939. Cell colors (blue–white–red gradient) correspond to the binary logarithm of the ratio of the expression level in a current sample to the average level across all the samples (per each transcript). Blue, expression level is below the average; red, above the average.

**Figure 5 ijms-23-12837-f005:**
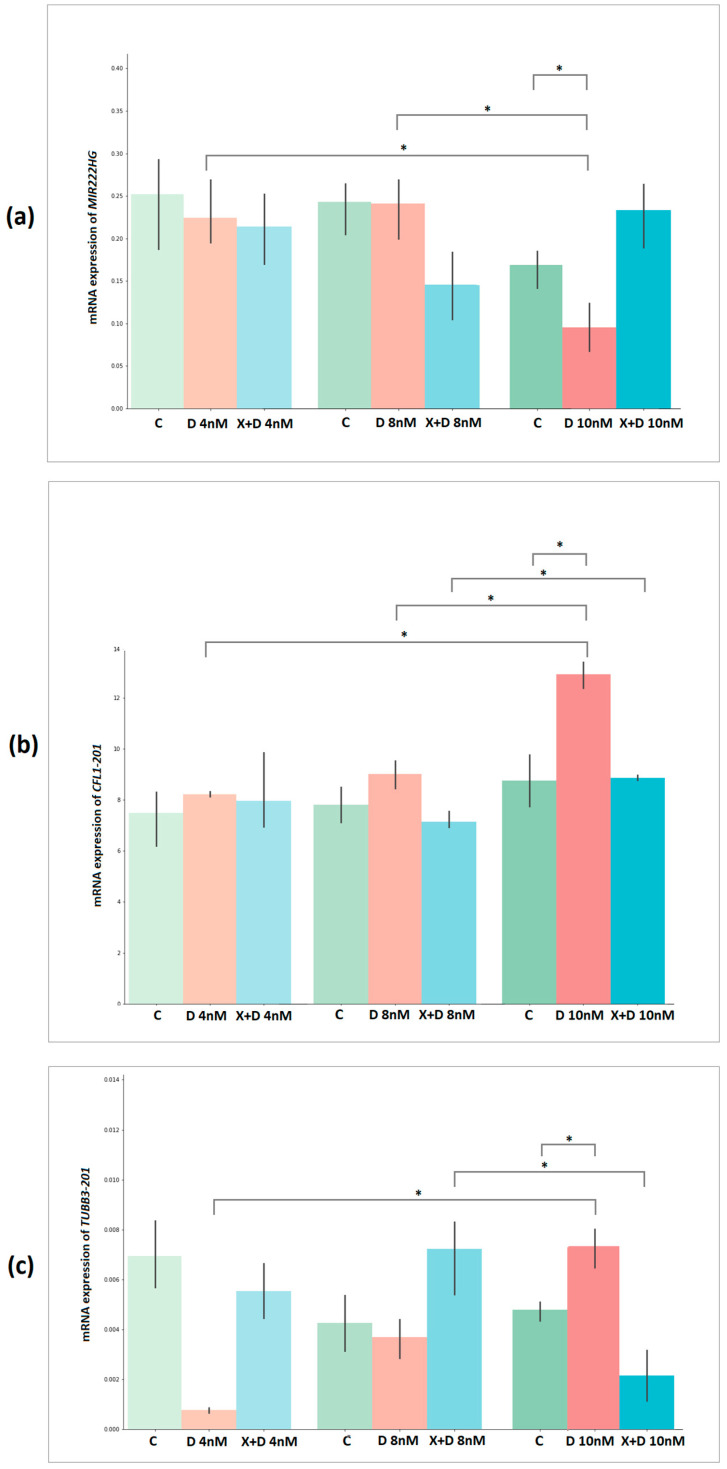
Bar plots showing relative expression results for (**a**) MIR222HG, (**b**) CFL1-201 and (**c**) TUBB3-201 transcripts between experimental groups of PC3 cells. The *y*-axis represents the 2^−ΔCT^ values. C—control, D—docetaxel, X+D—XAV939+Docetaxel. *—significant alteration.

**Figure 6 ijms-23-12837-f006:**
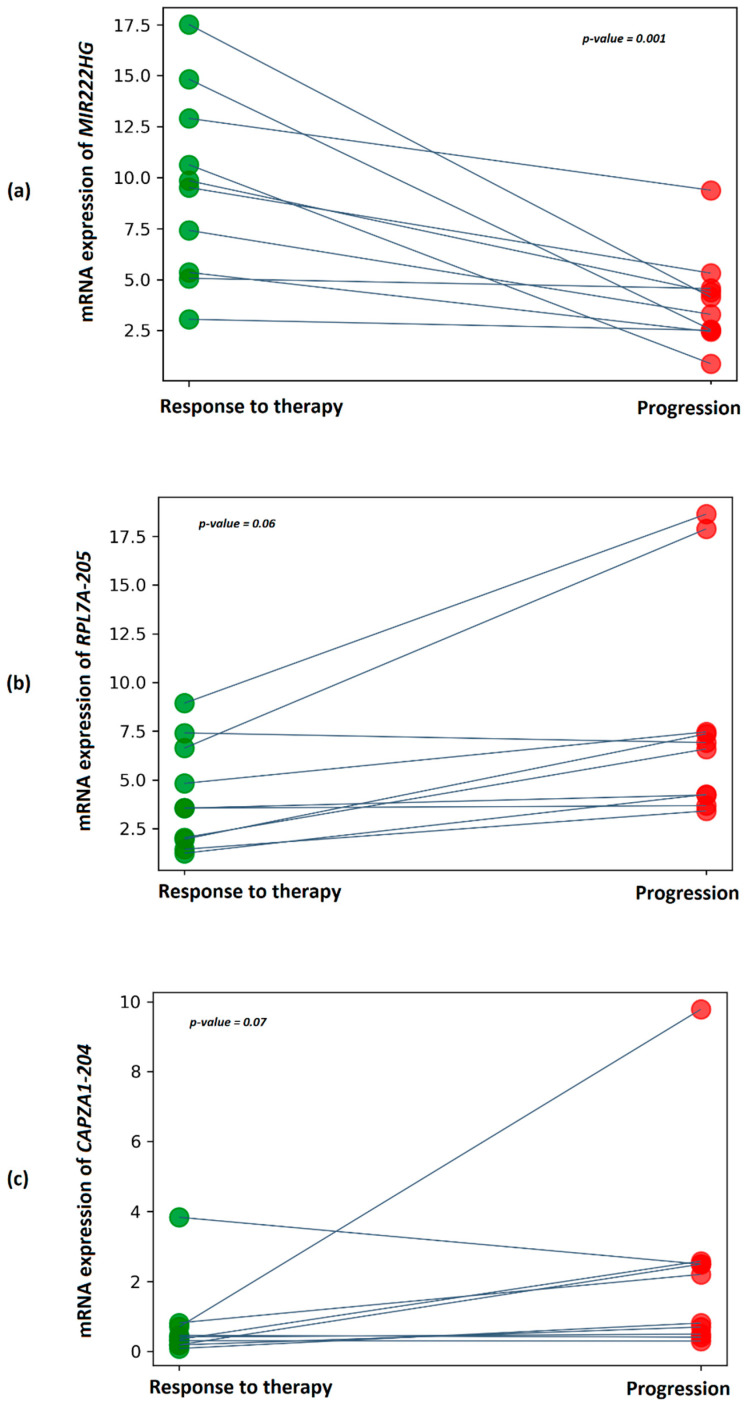
Dot plots showing relative expression results for (**a**) *MIR222HG*, (**b**) *RPL7A-205*, and (**c**) *CAPZA1-204* between docetaxel response periods and progression based on plasma exosome samples from patients with CRPC. The *y*-axis represents the 2^-ΔCT values. The gray line indicates the direction of relative expression of the transcript between periods for each patient.

**Table 1 ijms-23-12837-t001:** List of statistically significant pathways associated with different periods of PC3 cells docetaxel treatment, according to the Gene Ontology Biological Process 2021 database.

Pathway ID	Pathway Name	ES	NES	*p*-Value
	*Docetaxel 4 nM vs. Control 1*			
GO:0001819	positive regulation of cytokine production	0.51	1.45	1.00 × 10^−2^
GO:0051896	regulation of protein kinase B signaling	−0.64	−1.37	2.00 × 10^−2^
	*Docetaxel 8 nM vs. Control 2*			
GO:0071357	cellular response to type I interferon	0.63	1.60	1.00 × 10^−2^
GO:0071346	cellular response to interferon-gamma	0.65	1.60	1.00 × 10^−2^
GO:0006954	inflammatory response	0.63	1.58	1.00 × 10^−2^
GO:0140546	defense response to symbiont	0.59	1.58	1.00 × 10^−2^
GO:0051607	defense response to virus	0.57	1.54	1.00 × 10^−2^
GO:0071427	mRNA-containing ribonucleoprotein complex export from nucleus	−0.37	−1.25	1.00 × 10^−2^
GO:0042254	ribosome biogenesis	−0.41	−1.33	1.00 × 10^−2^
GO:0090630	activation of GTPase activity	−0.48	−1.39	1.00 × 10^−2^
GO:0000070	mitotic sister chromatid segregation	−0.39	−1.39	1.00 × 10^−2^
GO:0016072	rRNA metabolic process	−0.41	−1.41	1.00 × 10^−2^
GO:0140014	mitotic nuclear division	−0.48	−1.43	1.00 × 10^−2^
GO:0006260	DNA replication	−0.41	−1.45	1.00 × 10^−2^
GO:0006364	rRNA processing	−0.45	−1.52	1.00 × 10^−2^
GO:0048015	phosphatidylinositol-mediated signaling	−0.67	−1.64	1.00 × 10^−2^
GO:0034470	ncRNA processing	−0.46	−1.78	1.00 × 10^−2^
GO:0034470	cytokine-mediated signaling pathway	0.49	1.39	1.00 × 10^−2^
GO:0060337	type I interferon signaling pathway	0.63	1.59	1.00 × 10^−2^
GO:0034341	response to interferon-gamma	0.64	1.47	1.00 × 10^−2^
GO:0045071	negative regulation of viral genome replication	0.54	1.37	2.00 × 10^−2^
GO:0050708	regulation of protein secretion	0.60	1.46	4.00 × 10^−2^
GO:0098662	inorganic cation transmembrane transport	−0.48	−1.43	4.00 × 10^−2^
GO:0071345	cellular response to cytokine stimulus	0.46	1.31	4.00 × 10^−2^
GO:0072594	establishment of protein localization to organelle	−0.47	−1.36	4.00 × 10^−2^
	*Docetaxel 10 nM vs. Control 3*			
GO:0006954	inflammatory response	0.65	1.65	1.00 × 10^−2^
GO:0060337	type I interferon signaling pathway	0.60	1.60	1.00 × 10^−2^
GO:0050727	regulation of inflammatory response	0.53	1.49	1.00 × 10^−2^
GO:0032675	regulation of interleukin-6 production	0.59	1.43	2.00 × 10^−2^
GO:0048523	negative regulation of cellular process	0.45	1.39	2.00 × 10^−2^
GO:0098662	inorganic cation transmembrane transport	0.63	1.58	2.00 × 10^−2^
GO:0071357	cellular response to type I interferon	0.60	1.52	4.00 × 10^−2^
GO:0060333	interferon-gamma-mediated signaling pathway	0.63	1.51	4.00 × 10^−2^
GO:0001818	negative regulation of cytokine production	0.56	1.41	4.00 × 10^−2^
GO:0070555	response to interleukin-1	0.63	1.49	4.00 × 10^−2^
GO:0019221	cytokine-mediated signaling pathway	0.44	1.40	4.00 × 10^−2^
GO:0006915	apoptotic process	0.50	1.41	4.00 × 10^−2^
GO:0071346	cellular response to interferon-gamma	0.62	1.61	4.00 × 10^−2^
GO:0032101	regulation of response to external stimulus	0.57	1.45	5.00 × 10^−2^
	*Docetaxel 8 nM vs. Docetaxel 10 nM*			
GO:0071357	cellular response to type I interferon	0.69	3.3	1.00 × 10^−2^
GO:0060337	type I interferon signaling pathway	0.69	3.2	1.00 × 10^−2^
GO:0060333	interferon-gamma-mediated signaling pathway	0.56	2.39	1.38 × 10^−3^
GO:0045824	negative regulation of innate immune response	0.57	2.32	4.50 × 10^−3^
GO:0006811	ion transport	0.59	2.32	4.95 × 10^−3^
GO:0071346	cellular response to interferon-gamma	0.46	2.2	1.55 × 10^−2^
GO:0008203	cholesterol metabolic process	0.49	2.2	1.62 × 10^−2^
GO:0034341	response to interferon-gamma	0.52	2.17	1.73 × 10^−2^
GO:0071357	cellular response to type I interferon	0.69	3.3	1.00 × 10^−2^
GO:0060337	type I interferon signaling pathway	0.69	3.2	1.00 × 10^−2^
GO:0060333	interferon-gamma-mediated signaling pathway	0.56	2.39	1.38 × 10^−3^
GO:0045824	negative regulation of innate immune response	0.57	2.32	4.50 × 10^−3^
GO:0006811	ion transport	0.59	2.32	4.95 × 10^−3^
GO:0071346	cellular response to interferon-gamma	0.46	2.2	1.55 × 10^−2^
GO:0008203	cholesterol metabolic process	0.49	2.2	1.62 × 10^−2^
GO:0034341	response to interferon-gamma	0.52	2.17	1.73 × 10^−2^
GO:0048878	chemical homeostasis	0.54	2.13	2.09 × 10^−2^
GO:0014070	response to organic cyclic compound	0.70	2.13	1.00 × 10^−2^
GO:0019221	cytokine-mediated signaling pathway	0.32	2.12	2.26 × 10^−2^
GO:0060338	regulation of type I interferon-mediated signaling pathway	0.56	2.1	2.54 × 10^−2^
GO:0050727	regulation of inflammatory response	0.4	2.08	2.91 × 10^−2^
GO:0034599	cellular response to oxidative stress	0.38	2.07	2.95 × 10^−2^
GO:0006006	glucose metabolic process	0.48	2.03	3.56 × 10^−2^
GO:0001960	negative regulation of cytokine-mediated signaling pathway	0.49	2.03	3.66 × 10^−2^
GO:2000377	regulation of reactive oxygen species metabolic process	0.47	2.02	3.66 × 10^−2^
GO:0050728	negative regulation of inflammatory response	0.43	2.02	3.67 × 10^−2^
GO:0010952	positive regulation of peptidase activity	0.55	2.01	3.67 × 10^−2^
GO:0014070	response to organic cyclic compound	0.6	2.03	3.71 × 10^−2^
GO:0055092	sterol homeostasis	0.51	2.01	3.71 × 10^−2^
GO:0006695	cholesterol biosynthetic process	0.52	1.96	4.73 × 10^−2^
GO:0019216	regulation of lipid metabolic process	0.42	1.97	4.83 × 10^−2^
GO:0002474	antigen processing and presentation of peptide antigen via MHC class I	0.5	1.96	4.88 × 10^−2^
GO:0050728	negative regulation of inflammatory response	0.50	1.79	1.00 × 10^−2^
GO:0034599	cellular response to oxidative stress	0.36	1.48	1.00 × 10^−2^

**Table 2 ijms-23-12837-t002:** Statistically significant pathways associated with the docetaxel resistance of PC3 and DU-145 cells, according to the Gene Ontology Biological Process 2021 database.

Dataset	Parameters	GO:0014070	GO:0050728	GO:0034599
PC3	ES	0.70	0.50	0.49
	NES	2.14	1.79	1.47
	*p*-value	1.00 × 10^−2^	1.00 × 10^−2^	3.13 × 10^−2^
PC3-SC	ES	0.58	0.44	0.38
	NES	1.91	1.83	1.55
	*p*-value	1.00 × 10^−2^	1.00 × 10^−2^	2.30 × 10^−2^
DU145-SC	ES	−0.56	0.49	0.66
	NES	−1.87	1.65	2.55
	*p*-value	1.00 × 10^−2^	2.44 × 10^−2^	1.00 × 10^−2^

**Table 3 ijms-23-12837-t003:** List of statistically significant pathways, based on the list of DE genes between PC3 cells treated with XAV939+DOX 8 nM and XAV939+10 nM, according to the Gene Ontology Biological Process 2021 database.

Pathway ID	Pathway Name	ES	NES	FDR
GO:0045087	innate immune response	0.58	2.14	4.53 × 10^−2^
GO:0010467	gene expression	−0.34	−1.79	4.01 × 10^−2^
GO:0006396	RNA processing	−0.5	−1.88	1.59 × 10^−2^
GO:0000184	nuclear-transcribed mRNA catabolic process nonsense-mediated decay	−0.48	−2.02	6.09 × 10^−3^
GO:0042254	ribosome biogenesis	−0.45	−2.04	4.43 × 10^−3^
GO:0006334	nucleosome assembly	−0.62	−2.06	3.00 × 10^−3^
GO:0045047	protein targeting to ER	−0.49	−2.09	2.74 × 10^−3^
GO:0065004	protein–DNA complex assembly	−0.52	−2.11	2.45 × 10^−3^
GO:0006281	DNA repair	−0.41	−2.12	2.11 × 10^−3^
GO:0016072	rRNA metabolic process	−0.54	−2.14	2.28 × 10^−3^
GO:0002181	cytoplasmic translation	−0.51	−2.16	1.87 × 10^−3^
GO:0006613	cotranslational protein targeting to membrane	−0.58	−2.16	2.06 × 10^−3^
GO:0034728	nucleosome organization	−0.59	−2.19	1.52 × 10^−3^
GO:0031497	chromatin assembly	−0.62	−2.27	8.56 × 10^−4^
GO:0006614	SRP-dependent cotranslational protein targeting to membrane	−0.61	−2.29	9.79 × 10^−4^
GO:0006364	rRNA processing	−0.55	−2.29	1.14 × 10^−3^
GO:0022618	ribonucleoprotein complex assembly	−0.61	−2.3	1.37 × 10^−3^
GO:0006397	mRNA processing	−0.53	−2.37	1.00 × 10^−4^
GO:0000377	RNA splicing via transesterification reactions with bulged adenosine	−0.57	−2.43	1.00 × 10^−4^
GO:0034470	ncRNA processing	−0.55	−2.46	1.00 × 10^−4^
GO:0000398	mRNA splicing via spliceosome	−0.57	−2.46	1.00 × 10^−4^

**Table 4 ijms-23-12837-t004:** List of statistically significant DE lncRNAs associated with the development of resistance to docetaxel.

lncRNA	Data Set	Log2FC	FDR/*p*-Value QLF
*AL157938.2*	PC3	-	-
	PC3-CS	−1.99	1.12 × 10^−4^
	DU-145 SC	−5.08	1.23 × 10^−9^
	Plasma exosomes	−3.49	2.15 × 10^−2^
*LINC02582*	PC3	-	-
	PC3-CS	−1.2	5.07 × 10^−10^
	DU-145 SC	−2.43	4.24 × 10^−6^
	Plasma exosomes	−1.89	1.73 × 10^−2^
*SNHG1*	PC3	−0.61	9.97 × 10^−10^
	PC3-CS	−1.08	9.14 × 10^−3^
	DU-145 SC	−1.45	2.21 × 10^−5^
	Plasma exosomes	-	-
*KCNQ1OT1*	PC3	−0.38	8.22 × 10^−3^
	PC3-CS	−2.05	1.05× 10^−18^
	DU-145 SC	-	-
	Plasma exosomes	−1.83	4.30 × 10^−2^
*MIR222HG*	PC3	−0.88	9.94 × 10^−5^
	PC3-CS	−2.15	7.97 × 10^−5^
	DU-145 SC	-	-
	Plasma exosomes	−3.57	9.22 × 10^−3^

**Table 5 ijms-23-12837-t005:** List of statistically significant DE transcripts associated with the development of resistance to docetaxel treatment on PC3 cells.

Transcript ID	Transcript Name	Transcript Type	Log2FC	QLF FDR
ENST00000429677	*PRSS3-204*	protein_coding	0.74	2.54 × 10^−3^
ENST00000331825	*FTL-201*	protein_coding	0.8	1.83 × 10^−6^
ENST00000264832	*ICAM1-201*	protein_coding	1.04	5.92 × 10^−6^
ENST00000558131	*RPL28-205*	protein_coding	0.68	5.74 × 10^−5^
ENST00000683756	*AMBRA1-213*	protein_coding	4.86	2.19 × 10^−2^
ENST00000252809	*GDF15-201*	protein_coding	2.32	1.66 × 10^−6^
ENST00000380394	*RPS6-204*	protein_coding	0.82	3.04 × 10^−3^
ENST00000259874	*IER3-201*	protein_coding	0.73	8.69 × 10^−4^
ENST00000458500	*RPL10-210*	protein_coding	0.45	1.13 × 10^−3^
ENST00000339647	*UBC-201*	protein_coding	0.78	5.25 × 10^−6^
ENST00000389805	*SQSTM1-202*	protein_coding	1.45	1.57 × 10^−7^
ENST00000302754	*JUNB-201*	protein_coding	0.89	2.13 × 10^−2^
ENST00000308162	*CFL1-201*	protein_coding	1.42	4.91 × 10^−9^
ENST00000340384	*TUBB4B-201*	protein_coding	0.71	2.54 × 10^−5^
ENST00000327892	*TUBB-205*	protein_coding	0.51	3.48 × 10^−2^
ENST00000368719	*S100A6-201*	protein_coding	0.94	5.39 × 10^−6^
ENST00000315491	*TUBB3-201*	protein_coding	1.25	2.93 × 10^−4^

**Table 6 ijms-23-12837-t006:** Protein products of identified mRNA isoforms associated with the development of resistance to docetaxel in the PC3 cells. The annotation was obtained from the UniProt database.

Transcript Name	Protein Stable ID	UniProt Swiss-Prot ID	UniProt TrEMBL ID	Protein Length
*PRSS3-204*	ENSP00000401828	P35030	-	304
*FTL-201*	ENSP00000366525	P02792	A0A384MDR3	175
*ICAM1-201*	ENSP00000264832	P05362	A0A384MEK5	532
*RPL28-205* *AMBRA1-213*	ENSP00000453285 ENSP00000508322	- Q9C0C7	H0YLP6 -	89 1298
*GDF15-201*	ENSP00000252809	Q99988	-	308
*RPS6-204*	ENSP00000369757	P62753	A2A3R6	249
*IER3-201*	ENSP00000259874	P46695	A0A1U9X7X2	156
*RPL10-210*	ENSP00000395025	-	A6QRI9	181
*UBC-201*	ENSP00000344818	P0CG48	-	685
*SQSTM1-202*	ENSP00000374455	Q13501	-	440
*JUNB-201*	ENSP00000303315	P17275	Q5U079	347
*CFL1-201*	ENSP00000309629	P23528	V9HWI5	166
*TUBB4B-201*	ENSP00000341289	P68371	-	445
*TUBB-205*	ENSP00000339001	P07437	Q5SU16	444
*S100A6-201*	ENSP00000357708	P06703	-	90
*TUBB3-201*	ENSP00000320295	Q13509	-	450

**Table 7 ijms-23-12837-t007:** Differential expression of *PRSS3, FTL, UBC, SQSTM1* and *CFL1* genes.

Gene ID	Gene Name	PC3 Log2FC	QLF FDR	PC3-SC Log2FC	QLF FDR	DU145-SC Log2FC	DU145-SC QLF FDR
ENSG00000010438	*PRSS3*	0.46	2.38 × 10^−7^	0.48	1.99 × 10^−2^	4.67	3.71 × 10^−11^
ENSG00000087086	*FTL*	0.58	2.00 × 10^−23^	0.81	8.52 × 10^−10^	-	-
ENSG00000150991	*UBC*	0.49	1.63 × 10^−17^	0.54	1.83 × 10^−3^	-	-
ENSG00000161011	*SQSTM1*	1.16	1.51 × 10^−66^	1.11	1.83 × 10^−6^	-	-
ENSG00000172757	*CFL1*	0.66	2.93 × 10^−30^	0.74	3.90 × 10^−9^	-	-

**Table 8 ijms-23-12837-t008:** Differential expression of transcripts potentially associated with the development of resistance to docetaxel treatment when PC3 cells are treated with XAV939+Docetaxel combination.

Transcript ID	Transcript Name	DOX Log2FC	XAV939+DOX Log2FC
ENST00000429677	*PRSS3-204*	0.74	−0.1
ENST00000331825	*FTL-201*	0.8	0.39
ENST00000264832	*ICAM1-201*	1.04	0.9
ENST00000558131	*RPL28-205*	0.68	0.18
ENST00000683756	*AMBRA1-213*	4.86	0.08
ENST00000252809	*GDF15-201*	2.32	1.7
ENST00000380394	*RPS6-204*	0.82	−0.12
ENST00000259874	*IER3-201*	0.73	0.54
ENST00000458500	*RPL10-210*	0.45	−0.13
ENST00000339647	*UBC-201*	0.78	0.25
ENST00000389805	*SQSTM1-202*	1.45	1.14
ENST00000302754	*JUNB-201*	0.89	−0.02
ENST00000308162	*CFL1-201*	1.42	0.05
ENST00000340384	*TUBB4B-201*	0.71	0.04
ENST00000327892	*TUBB-205*	0.51	−0.01
ENST00000368719	*S100A6-201*	0.94	0.33
ENST00000315491	*TUBB3-201*	1.25	0

**Table 9 ijms-23-12837-t009:** List of statistically significant DE transcripts associated with the development of resistance to docetaxel treatment based on plasma exosomes of patients with CRPC.

Transcript ID	Transcript Name	Transcript Type	Log2FC	QLF *p*-Value	r_s_	r_s_ *p*-Value
ENST00000468019	*RPL7A-205*	lncRNA	4.86	2.19 × 10^−3^	0.53	1.59 × 10^−2^
ENST00000598681	*NOP53-207*	lncRNA	2.97	3.55 × 10^−2^	0.46	4.14 × 10^−2^
ENST00000476936	*CAPZA1-204*	lncRNA	3.19	2.37 × 10^−2^	0.64	2.57 × 10^−3^
ENST00000508832	*MALAT1-201*	lncRNA	2.84	3.39 × 10^−2^	0.48	3.18 × 10^−2^

**Table 10 ijms-23-12837-t010:** Results of a *t*-test based on the relative expression of *MIR222HG*, *CFL1-201* and *TUBB3-201* between experimental groups of PC3 cells.

Groups	*MIR222HG*	*p*-Value	*CFL1-201*	*p*-Value	*TUBB3-201*	*p*-Value
DOX 4 nM vs. DOX 8 nM	−0.32	7.62 × 10^−1^	−2.46	6.96 × 10^−2^	−2.21	9.16 × 10^−2^
DOX 4 nM vs. DOX 10 nM	4.45	1.13 × 10^−2^	−16.15	8.61 × 10^−5^	−14.52	1.31 × 10^−4^
DOX 8 nM vs. DOX 10 nM	3.09	3.66 × 10^−2^	−9.28	7.49 × 10^−4^	−2.62	5.88 × 10^−2^
Control 1 vs. DOX 4 nM	0.46	6.69 × 10^−1^	−1.12	3.27 × 10^−1^	1.83	1.41 × 10^−1^
Control 2 vs. DOX 8 nM	0.04	9.70 × 10^−1^	−2.38	7.63 × 10^−2^	0.29	7.86 × 10^−1^
Control 3 vs. DOX 10 nM	3.46	2.58 × 10^−2^	−6.38	3.09 × 10^−3^	−5.06	7.16 × 10^−3^
XAV939+DOX 8 nM vs. 10 nM	−0.91	4.13 × 10^−1^	−8.36	1.12 × 10^−3^	4.68	9.45 × 10^−3^

**Table 11 ijms-23-12837-t011:** Clinical and pathological characteristics of patients with CRPC included in the study.

Patients	Age	Gleason Score	PSA at Diagnosis CRPC, ng/mL	Radionuclide Study of the Skeletal System
Pat1	66	9 (5 + 4)	3000	multiple bone metastasis
Pat2	68	9 (5 + 4)	52	multiple bone metastasis
Pat3	71	8 (4 + 4)	1200	bone metastasis
Pat4	68	8 (4 + 4)	1900	bone metastasis
Pat5	61	8 (4 + 4)	23	multiple bone metastasis
Pat6	68	8 (4 + 4)	124	bone metastasis
Pat7	70	8 (4 + 4)	72	bone metastasis
Pat8	73	8 (4 + 4)	2950	bone metastasis
Pat9	66	8 (4 + 4)	334	bone metastasis
Pat10	69	9 (4 + 5)	37	bone metastasis
Pat11	66	-	521	bone metastasis

## Data Availability

All data generated or analyzed during this study are available within the article or upon request from the corresponding author.

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
