# Peer review of "Docetaxel Resistance in Castration-Resistant Prostate Cancer: Transcriptomic Determinants and the Effect of Inhibiting Wnt/β-Catenin Signaling by XAV939"

_ijms, 2022, doi:10.3390/ijms232112837_

Round 1

Reviewer 1 Report

Despite advances in prostate cancer diagnosis and management, mortality from prostate cancer remains high especially castration-resistant prostate cancer (CRPC). This manuscript solves two problems:  docetaxel resistance and combined effect with XAV-939 on maintaining the sensitivity of tumor cells to chemotherapy using transcriptome analysis both in PC3 cell lines and plasma exosome samples from patients with CRPC.  Finally, they identify a 29 number of mRNA isoforms with high predictive potential that can be considered as promising 30 markers of response to docetaxel. In general, the article gives a good impression, it is performed at a good scientific level. I have a few small remarks: 1. in Abstract it is necessary to decipher the compound XAV-939 as a tankyrase inhibitor. 2. In the Discussion section, there is no author’s opinion why the identified genes do not match the liquid biopsy data. In response to these comments, the article can be recommended for publication.

Author Response

Dear Reviewer,
Thank you very much for reviewing our research.
Answers to remarks:
1. Transcript for XAV-939 added to abstract.
2. The results of our study were supplemented by comparing the lists of differentially expressed genes between the data of blood plasma exosomes, cell samples, as well as an additional set of transcriptomic data. For lncRNA MIR222HG, which was detected when comparing all data sets, statistical significance was shown based on the results of the qPCR validation.

Reviewer 2 Report

The manuscript by E Pudova et al presents interesting data on gene expression in prostate cancer following exposure to docetaxel in the presence and absence of the inhibitor XAV939.  However, multiple issues should be addressed before the manuscript would be considered suitable for publication:

-All panels in Figure 1B are labeled the same.  As a result, it is difficult to determine what the difference is between these treatments (especially when  the responses vary from panel to panel).  The figure should be relabeled to better explain the data in this panel. 

-Figure 3 is difficult to read and interpret in its current form.  A better version of the image with enhanced text should be included in the manuscript.

-The authors note there is toxicity following administration of XAV939 to PC-3 cells. However no cell count or viability data to support this claim is included in the manuscript.  The authors should insert data from cell viability assays or cell count assays to support this claim and further substantiate the concentration of XAV939 used throughout the study.

-The authors attempt to connect the data obtained in the patient samples with the data obtained in the PC3 cells.  However, based on the data presented, the patient samples appear to come from patients who have tumors that express the androgen receptor (AR).  Since the PC3 cells contain little to no AR, it is difficult to make a correlation/comparison between the two sets of data.  The authors should discuss this discrepancy in the text.

Minor Revisions:

For consistency between the data in the main paper and that in the supplemental tables, the commas in tables 1,2,4,5 and 6 should be changed to periods where appropriate.   

Line 29 –Text should be modified so it reads “in the treatment of CRPC; moreover, they suggest...”

Line 46- The phrase “violation of cell mitosis” is not clear.  This portion of the sentence should be rephrased to enhance clarity.

Line 205 – Text should be modified so it reads “(significant rise of prostate specific antigen)”

Line 227-230- There are some repeated sentences in these lines.  Text should be revised to remove the redundancy.

Line 313 – Text should be modified so it reads “ …autophagosomes. The p62 protein …”.

Line 330 – Text should be modified so it reads “enrichment of biological pathways. However, when assessed in …”.

Author Response

Dear Reviewer,
Thank you very much for reviewing our research.
Answers to key remarks:
- Figure 1B (now Figure 2b) corrected
- Figure 3 (now Figure 4) corrected
In addition, the drawing files will be available for detailed review in the future.
- Added cell count data after three rounds of XAV939 treatment (Figure 2a)
- Patients with CRPC included in the study had previously undergone radical prostatectomy. This explanation has been included in the Materials and Methods section.

All minor comments have been corrected.

Reviewer 3 Report

The manuscript presents profound transcriptomic studies of changes in biochemical processes in cultured tumor cells of the castration-resistant prostate cancer while the docetaxel treatment, formation of the resistance to this drug and its overcome by the Wnt/β-catenin signaling pathway inhibitor XAV939. Though some aspects of the docetaxel action and resistance formation in tumor cells were revealed before the authors' studies, they presented only a fragmented biochemical landscape, while this manuscript allows a much wider and deeper view on the processes under study. In addition, changes in patient exosomal lncRNA while docetaxel resistance formation were studied that may lay a basement for minimally invasive diagnostics. The presented data are of value and interest for the readers of IJMS, specialists in the corresponding field. Methodology of the research is well described.  

However, certain notes should be made just to improve the quality of presentation. 
1) Line205 - the word "ise"-what does it mean? Suppose it is a typing error.
2) Line 281- "co-culture": there should be an explanation, what cells were co-cultured ,otherwise it remains unclear. 
3) Lines 290-291 -the enzymes listed in these lines, are NOT involved in LIPOLYSIS, but in other processes of lipid metabolism.
4)Line 317 - abbreviation BPH is not deciphered.
5) Line 403 - filter pore size 0.8 m -is it right?
6) XAV-939 is not a common designation, the common one for this Wnt/β-catenin signaling pathway inhibitor is XAV939, the use of which will make a corresponding paper search more easily. 

Regarding the data presentation in the Supplementary materials, it would be better to mark statistically significant changes in tables, e.g. by different colours, or in another manner, just to make such changes more clearly visible.   

Author Response

Dear Reviewer,
Thank you very much for reviewing our research.
All highlighted typos and inaccuracies in the manuscript have been corrected.

Reviewer 4 Report

The manuscript essentially described subsequent, or sequential, changes in mRNA expression after treatment with taxanes and a WNT signaling modulator. The use of extracellular vesivcle for this analysis, in addition to the "standard" RNAseq, is a very positive twist in the otherwise simple study design, and much appreciated. However, the EVs were isolated from patients, and we have no idea if and how they should be related to PC3 cells, and how (and if) data generated on these completely different settings can or should be compared?

The authors have exclusively focused on PC3 cells - which are maybe untypical for PrCa because these cells have lost all traces of AR expression. Treating these cells with taxanes is of yourse a possibility - and will certainly result in more or less specific responses. If and how they are related to prostate cancer progression, in particular, to CRPC, is another question. Especially, if these should be related in any way to AR signaling and activities, including AR-regulated genes (like KLK3/PSA; which arent expressed in PC3 cells dues to the lack of AR expression). 

A list of transcripts follows later, as Table 2 and 3. It remains unclear to me why the official gene symbols used in Table 3 have been modified. Why has there always a "-204" or "201" added to the gene name? I dont think this has been explained properly. How do you get from the number to the corresponding isoform? And is the expression of the gene also different as such, not just for one of the isoforms? Why not show a few PCRs to validate differential expression, that would be a low-level and almost symbolic control, but it appears necessary to gain trust in the data. 

Apart of these specific questions, it also remains to be mentioned that results from just one (1) single cell line are difficult to interpret, if they are not validated by at least one or several other cell lines. It should therefore be asked (for all medium- to high impact factor journals) to back up the data from 1 cell line with another, independent set based on another line. Or, if you can, patient- or clinical tissues. 

For example, how do we know that the very prominent interferone alpha & beta or gamma response (Table 1) observed after taxane treatments isnt a PC3-specific trait; we will not be able to judge if this may also happen in other PrCa cell lines that more or less mimic CRPC. 

Instead of the multiple volcano plots shown in Fig. 1, it should be considered to show heatmaps, but in a fashion that would allow the reader to identify interesting gene names that are differentially expressed in the successive treatment schedule. 

We also do not know why pathways that involve microtiubuli turnover arent more prominently affected ? This paradoxon is only briefly discussed in a sort paragraph (lines 162 - 166) but not in an explanatory fashion. Its not clearm for example, how WNT pathway inhibition would or should affect mitotic spindle formation, and how this is affected by the taxanes. Where is the functional/mechanistic overlap? Why doont tubulin-related GO pathways already pop up with the 1st taxane treatments (where you would expect them to be most prominently). 

Generally, there is much more of a focus on pathways, and gene ontology categories, than on single candidate genes and therefore, functional issues that could be tested and validated.  Which are the most prominently affected genes? Is their expession also altered at the protein level? (IN line 184, finally one such candidate is mentioned, TUBB3), which is of course one of the "usual suspects".  But no further validation. 

ON Fig. 3, finally transcripts (not just GO categories) are discussed, but the gene names/symbols are so small that I can hardly deipher anything. Fig. 3 needs to be revised and significantly improved. 

Altogether, the study is not "rounded up", there are bits of pieces that are isolated and difficult to connect (e.g., PC3/EV data). There is no precedence in which such detailed analyses may have been done. There is also no validation whatsoever for any of the multiple findings, which are at different hierrchical layers 

Small issues:

there are a number of terms, that are just incorrect, and need to be changed. For example (line 46), what does "violation of cell mitosis" mean? (I suppose, cell cycle arrest?). 

Author Response

Dear Reviewer,
Thank you very much for reviewing our research. Your comments have helped us improve the quality of our study results.

We would like to respond to comments:
-Plasma exosome samples from patients with CRPC who had previously undergone radical prostatectomy and had multiple bone metastases were included in the study. The PC3 prostate cancer cell line is the closest model match to our patient sample. It is also worth noting that the PC3 cell line is classically used as a model for advanced PCa in numerous studies evaluating various drugs, including chemotherapy drugs;
- The names of the gene transcripts were presented in tables according to the information in the Ensemble database;
- The results presented by us were supplemented by the data set of the PRJNA589746 project. Data includes docetaxel sensitive and resistant PC3 (PC3-SC) and DU-145 (DU145-SC) cell lines. In addition, we validated several transcripts by quantitative PCR, and confirmed the statistical significance of the results.
- Figures and inaccuracies in the text have been corrected.

Round 2

Reviewer 2 Report

The authors have addressed the areas of concern noted in the initial review.